# RELATION-BASED GENERALIZED ZERO-SHOT CLASSIFICATION WITH THE DOMAIN DISCRIMINATOR ON THE SHARED REPRESENTATION

## ABSTRACT

Generalized zero-shot learning (GZSL) is the task of predicting a test image from seen or unseen classes using pre-defined class-attributes and images from the seen classes. Typical ZSL models assign the class corresponding to the most relevant attribute as the predicted label of the test image based on the learned relation between the attribute and the image. However, this relation-based approach presents a difficulty: many of the test images are predicted as biased to the seen domain, i.e., the *domain bias problem*. Recently, many methods have addressed this difficulty using a synthesis-based approach that, however, requires generation of large amounts of high-quality unseen images after training and the additional training of classifier given them. Therefore, for this study, we aim at alleviating this difficulty in the manner of the relation-based approach. First, we consider the requirements for good performance in a ZSL setting and introduce a new model based on a variational autoencoder that learns to embed attributes and images into the shared representation space which satisfies those requirements. Next, we assume that the domain bias problem in GZSL derives from a situation in which embedding of the unseen domain overlaps that of the seen one. We introduce a discriminator that distinguishes domains in a shared space and learns jointly with the above embedding model to prevent this situation. After training, we can obtain prior knowledge from the discriminator of which domain is more likely to be embedded anywhere in the shared space. We propose combination of this knowledge and the relation-based classification on the embedded shared space as a mixture model to compensate class prediction. Experimentally obtained results confirm that the proposed method significantly improves the domain bias problem in relation-based settings and achieves almost equal accuracy to that of high-cost synthesis-based methods.

## 1 INTRODUCTION

The recent high performance of deep neural networks on image classification and object recognition depends greatly on whether one can obtain sufficiently labeled images of classes to predict. Nevertheless, it is difficult to do this in the real world because the number of existing classes is enormous. As long as human beings create or develop new objects, their number might continue to increase daily, thereby creating difficulty in obtaining labeled data of all classes to predict. In recent years, this difficulty led to great interest in zero-shot learning (ZSL) (Farhadi et al., 2009; Frome et al., 2013; Lampert et al., 2014; Xian et al., 2018a), which is training by a labeled set from certain classes called seen classes and then predicting completely unseen classes that are not included in the training set.

Usually, ZSL is accomplished by preparing pre-defined semantic representations of all classes, such as attributes, and learning the relation between images and the class-attributes (*relation-based* approach). Once it is learned from the training set, we can predict the labels of test examples from unseen classes by selecting the most relevant attributes on this relationship. However, it has been pointed out that this approach does not work for generalized zero-shot learning (GZSL), which is a more general setting where test samples can be from both seen and unseen classes (Chao et al., 2016). This is because all examples of the training set are obtained from the seen classes, so the

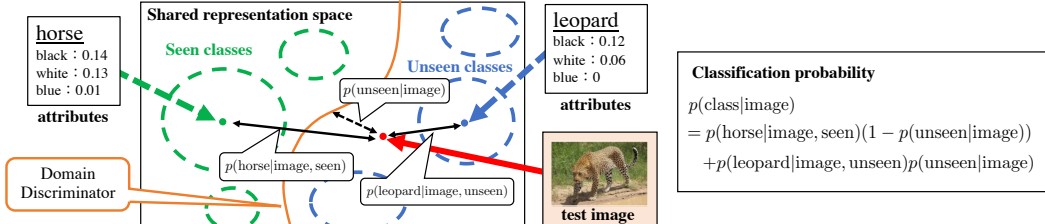

Figure 1: An overview of MCMAE-D. Images and attributes are embedded in the shared representation space by MCMAE inference models learned given the training set. We propose a model to distinguish different domains (seen or unseen) in space and to learn jointly with MCMAE. After learning, we can perform class prediction with a reduced bias toward the seen classes by combining relation-based classifier and domain discriminator as a mixture model, as shown in the equation on the right side.

data to be predicted as one of the unseen classes also has a strong relationship with the seen class attributes, resulting most of them assigned to one of the seen classes. In this paper, we refer to a domain as belonging to either seen or unseen, and call the problem that class prediction is biased to the seen domain as the *domain bias problem*.

To address this difficulty, recent works have taken an approach of learning a generative model that generates images from corresponding attributes, and of then training a classifier to predict classes from the generated synthesis images (Mishra et al., 2017; Verma & Rai, 2017; Xian et al., 2018b; Felix et al., 2018). The advantage of this synthesis-based approach is that samples of both domains are obtainable by generation, which contributes to alleviation of the domain bias problem. However, these *synthesis-based* methods require the generative model to generate numerous diverse and high-quality images for each class, including unseen ones, sufficient to classify with high-performance, which can be difficult and costly. In addition, this requires a classifier that classifies all classes from the generated images. On the other hand, conventional relation-based methods can make class predictions using only the learned image–attribute relation, not requiring enormous image generation and additional classifier training after training the model. Therefore, we address the following question in this paper: *Can we mitigate the domain bias problem of GZSL in the relation-based manner and achieve high performance?*

We first discuss the importance of the following requirements for good relation-based ZSL performance when embedding images and attributes in a shared representation space that satisfies the following requirements: images and attributes belonging to the same class must be in the same place (*modality invariance*); and different classes of samples must be separated in the shared space (*class separability*). To achieve such embedding, we propose *Modality-invariant and Class-separable Multimodal AutoEncoder (MCMAE)*, which is an extension of variational autoencoders (VAEs) (Kingma & Welling, 2013; Rezende et al., 2014). The objective of MCMAE is designed based on the two requirements presented above.

Next, we hypothesize that the domain bias problem results from a situation in which the unseen domain overlaps that of the seen one in the shared space. To address this point, we explicitly introduce a discriminator for separation of these two domains. This discriminator is trained jointly with MCMAE. After training, it gives the probability of a test image being in a given domain. In other words, it gives prior knowledge of the domain in the shared space. Based on this insight, we consider the class prediction probability as a soft combination of MCMAE classification and consider the domain discriminator as a mixture model (see Figure 1). Such combination-based classification has been proposed as a "gating" approach (Atzmon & Chechik, 2019). However, unlike this work, our method is able to train the entire model while retaining an end-to-end manner. We call this proposed approach as *MCMAE with the Domain discriminator (MCMAE-D)*.

The contribution of this research is the following. (1) We consider the requirements of the shared representation space to perform in the ZSL setting and introduce MCMAE as a model to learn the embedding of images and attributes into that space. (2) We also propose MCMAE-D, which combines MCMAE with the domain discriminator. The experiment results demonstrate that it greatly reduces the domain bias problem, thereby contributing to exceed the performances of the existing relation-based models greatly and to be equivalent to the state-of-the-art synthesis-based method.

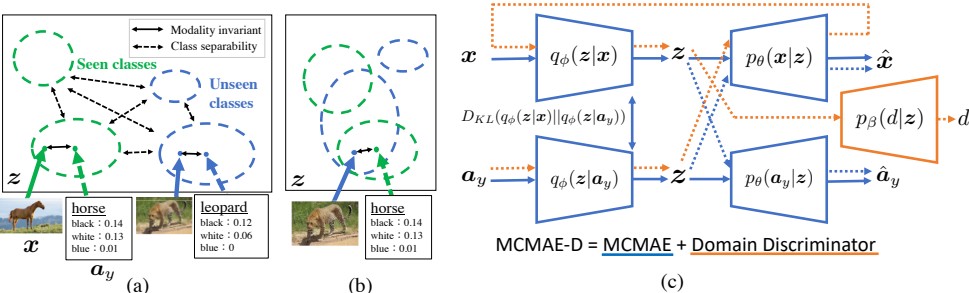

Figure 2: (a) Two requirements exist for achieving good performance in ZSL: modality invariant and class separability. (b) Failure of class separation between domains. The unseen domain overlaps with the seen domain. Therefore, all examples of the unseen classes might be predicted as one of the seen ones. (c) The network architecture of our proposed model, MCMAE-D.

## 2 PROBLEM FORMULATION: GENERALIZED ZERO-SHOT LEARNING

We assume that the dataset $\mathcal{D}_{tr} = \{\boldsymbol{x}_i, y_i\}_{i=1}^{N_{tr}}$ is given as the training set, where $\boldsymbol{x}_i \in \mathcal{X}$ is the input data, e.g. an image, and where $y_i \in \mathcal{Y}_s = \{1..., S\}$ is the corresponding label data.

The objective of ZSL is to learn the classifier using $\mathcal{D}_{tr}$ and to predict labels $\hat{y}_j \in \mathcal{Y}$ from the example $\boldsymbol{x}_j \in \mathcal{X}$ in the test set $\mathcal{D}_{ts} = \{\boldsymbol{x}_j, y_j\}_{j=1}^{N_{ts}}$. For the standard ZSL, it is assumed that the classes of the test set are completely unseen in the training set, which means that $\mathcal{Y} = \mathcal{Y}_u = \{S+1, ..., S+U\}$. Our goal is to train in the setting of GZSL, which includes both seen and unseen classes in the test set ($\mathcal{Y} = \mathcal{Y}_s \cup \mathcal{Y}_u$).

Furthermore, we assume that we have the class-attribute matrix $\boldsymbol{A} \in \mathbb{R}^{M \times (S+U)}$ as the semantic information of classes, where each column represents the $M$-dimensional attribute vector $\boldsymbol{a}_c \in \mathcal{A} = \mathbb{R}^M$ of each class $c = 1, ..., S + U$. Using this attribute vector, the training set $\mathcal{D}_{tr}$ can be replaced as $\{\boldsymbol{x}_i, \boldsymbol{a}_{y_i}\}_{i=1}^{N_{tr}}$.

Using the relation-based approach, the objective is changed to train the *compatibility function* of the input and attribute $F(\boldsymbol{x}, \boldsymbol{a}_y)$, which represents how the input and attribute are related: stronger relations have greater values. Once this function is learned from the training set, the classification probability can be expressed as

$$p(y = c|\boldsymbol{x}) = \frac{\exp\left(F(\boldsymbol{x}, \boldsymbol{a}_c)\right)}{\sum_{\hat{y} \in \mathcal{Y}} \exp\left(F(\boldsymbol{x}, \boldsymbol{a}_{\hat{y}})\right)}. \tag{1}$$

One can predict the class labels by choosing the one that maximizes this probability, i.e., $\hat{y} = \arg\max_{y \in \mathcal{Y}} p(y|\boldsymbol{x})$.

Additionally, we call seen and unseen ones as different *domains* and express it as a binary variable $d \in \{0, 1\}$, where $d = 0$ represents the seen domain and $d = 1$ represents the unseen one.

## 3 PROPOSED METHOD

### 3.1 RELATION-BASED CLASSIFICATION OF THE SHARED REPRESENTATION

In this study, the image $\boldsymbol{x}$ and the attribute $\boldsymbol{a}_y$ are regarded as different modalities. Also, the mapping $q_{\phi_x}(\boldsymbol{z}|\boldsymbol{x})$ and $q_{\phi_a}(\boldsymbol{z}|\boldsymbol{a}_y)$ embed them into the same space, i.e., the shared representation space $\boldsymbol{z}$. In this approach, the compatibility function can be expressed with the Kullback–Leibler (KL) divergence as

$$F_{\phi_{x,a}}(\boldsymbol{x}, \boldsymbol{a}_y) = -D_{KL}(q_{\phi_x}(\boldsymbol{z}|\boldsymbol{x})||q_{\phi_a}(\boldsymbol{z}|\boldsymbol{a}_y)), \tag{2}$$

where $\phi_{x,a}$ is a shorthand notation for $\phi_x$ and $\phi_a$. Therefore, learning the compatibility function corresponds to learning embeddings into the shared representation. Then, what kind of space embedding engenders good performance of ZSL and GZSL?

First, images and attributes belonging to the same class must be embedded in a nearby place in terms of KL divergence. It is clear that this embedding is a requirement for ZSL because, if not satisfied, the assumption on the relation-based methods is violated, rendering it impossible to predict corresponding classes from an input. Note that this requirement must also be generalized to unknown data, i.e., examples of unseen classes. That is, the shared representation must be *modality invariant* (see Figure 2(a)).

Second, examples embedded in the shared space by the mapping must be grouped by the same class and separated from other classes. If not satisfied (e.g., an area of one class overlaps with that of other classes), they may be misclassified as other classes. This can be rephrased that the embedded representation needs to be easily separated by some classifier, i.e., the shared representation must have *class separability* (Figure 2(a)).

*Class separability between domains* is particularly important for GZSL performance. Because only seen data are given during GZSL training, there is no clue to properly embed the unseen classes. Consequently, embedding of the unseen domain might overlap with those of the seen domain, and any input might be only to assigned to one of the more relevant seen classes (Figure 2(b)). *We hypothesize that the failure of class separability between domains is the underlying cause of the domain bias problem.*

## 3.2 MODALITY-INVARIANT AND CLASS-SEPARABLE MULTIMODAL AUTOENCODER

Given training data $(\boldsymbol{x}, \boldsymbol{a}_y)$, the simplest way to satisfy modality invariant is to maximize Eq. 2 over these data directly, but it is difficult to generalize to test data. For this study, we first introduce a shared representation learning method based on variational autoencoders (VAE) (Kingma & Welling, 2013; Rezende et al., 2014).

In VAE, the data $\boldsymbol{x}$ is assumed to be generated from a generative model $p_{\theta_x}(\boldsymbol{x}) = \int p_{\theta_x}(\boldsymbol{x}|\boldsymbol{z})p(\boldsymbol{z})d\boldsymbol{z}$ ($\theta_x$ is a learnable parameter), and learning is performed by maximizing the lower bound of the log marginal likelihood over the given data:

$$\mathcal{L}_{\theta_x,\phi_x}^{VAE}(\boldsymbol{x}) = E_{q_{\phi_x}(\boldsymbol{z}|\boldsymbol{x})}[\log p_{\theta_x}(\boldsymbol{x}|\boldsymbol{z})] - D_{KL}(q_{\phi_x}(\boldsymbol{z}|\boldsymbol{x})||p(\boldsymbol{z})), \tag{3}$$

where $q_{\phi_x}(\boldsymbol{z}|\boldsymbol{x})$ represents an approximate distribution of the true posterior $p_{\theta_x}(\boldsymbol{z}|\boldsymbol{x})$ and $\phi_x$ is a parameter. This approximate distribution, also called the inference model, can be regarded as an embedding from $\boldsymbol{x}$ into $\boldsymbol{z}$.

Although VAE learns to maximize Eq. 3 not under an exact generative distribution but under a finite training set, its representation is known to generalize well to unseen inputs. This capability is suitable for shared representation learning in ZSL, which needs to be generalized to the examples of the unseen classes.

To extend Eq. 3 to multimodal input of images and attributes, we first replace the prior in the regularization term from $p(\boldsymbol{z})$ to $q_{\phi_a}(\boldsymbol{z}|\boldsymbol{a}_y)$:

$$\mathcal{L}_{\theta_x,\phi_{x,a}}^{MAE}(\boldsymbol{x}, \boldsymbol{a}_y) = E_{q_{\phi_x}(\boldsymbol{z}|\boldsymbol{x})}[\log p_{\theta_x}(\boldsymbol{x}|\boldsymbol{z})] - D_{KL}(q_{\phi_x}(\boldsymbol{z}|\boldsymbol{x})||q_{\phi_a}(\boldsymbol{z}|\boldsymbol{a}_y)). \tag{4}$$

This makes regularization of the shared representation more relaxed because the replaced prior is learnable. In addition, maximizing this lower bound engenders learning to bring the two embeddings closer together explicitly.

Next, we introduce a model $p_{\theta_a}(\boldsymbol{a}_y|\boldsymbol{z})$ that discriminates attributes from the representation. By learning this model together with Eq. 4, one can add a constraint to the representation from which attributes can be successfully predicted:

$$\mathcal{L}_{\theta_{x,a},\phi_{x,a}}^{CMAE}(\boldsymbol{x}, \boldsymbol{a}_y) = \mathcal{L}_{\theta_x,\phi_{x,a}}^{MAE}(\boldsymbol{x}, \boldsymbol{a}_y) + E_{q_{\phi_x}(\boldsymbol{z}|\boldsymbol{x})}[\log p_{\theta_a}(\boldsymbol{a}_y|\boldsymbol{z})]. \tag{5}$$

Attributes are semantically distributed representations of classes, which engenders the class-separable representation. In addition, this equation can be interpreted as introducing an attribute generative model $p_\theta(\boldsymbol{a}_y|\boldsymbol{z})$. From such a perspective, this equation considers not only the reconstruction of an image, but also the "cross" reconstruction of the corresponding attribute from the image. This model is actually the same as the multimodal model known as PSE (Jiao et al., 2019), as described in Sec 4.2.

Furthermore, as discussed in Sec. 3.1, the shared representation must be modality-invariant. Therefore, we add an expected term at $q_{\phi_a}(\boldsymbol{z}|\boldsymbol{a}_y)$ in Eq. 5, which is a constraint by which the embedded representation is the same as long as it represents the same thing, irrespective of which modality inference model is used. Therefore, our objective is finally

$$
\begin{aligned}
\mathcal{L}_{\theta_{x,a},\phi_{x,a}}^{MCMAE}(\boldsymbol{x},\boldsymbol{a}_y) &= \mathcal{L}_{\theta_{x,a},\phi_{x,a}}^{CMAE}(\boldsymbol{x},\boldsymbol{a}_y) + E_{q_{\phi_a}(\boldsymbol{z}|\boldsymbol{a}_y)}[\log p_{\theta_{x,a}}(\boldsymbol{x},\boldsymbol{a}_y|\boldsymbol{z})] \\
&= E_{q_{\phi_x}(\boldsymbol{z}|\boldsymbol{x})}[\log p_{\theta_{x,a}}(\boldsymbol{x},\boldsymbol{a}_y|\boldsymbol{z})] + E_{q_{\phi_a}(\boldsymbol{z}|\boldsymbol{a}_y)}[\log p_{\theta_{x,a}}(\boldsymbol{x},\boldsymbol{a}_y|\boldsymbol{z})] \\
&\quad - D_{KL}(q_{\phi_x}(\boldsymbol{z}|\boldsymbol{x})||q_{\phi_a}(\boldsymbol{z}|\boldsymbol{a}_y)),
\end{aligned}
\tag{6}
$$

where $\log p_{\theta_{x,a}}(\boldsymbol{x},\boldsymbol{a}_y|\boldsymbol{z}) = \log p_{\theta_x}(\boldsymbol{x}|\boldsymbol{z}) + \log p_{\theta_a}(\boldsymbol{a}_y|\boldsymbol{z})$. All distributions are parameterized by deep neural networks. Moreover, we regard that the inference models as Gaussian and generative models as a Laplace distribution with a constant scale parameter. Therefore, the log-likelihood of the generative models is obtained by the negative absolute-difference loss.

This model considers all the requirements that the shared representation should satisfy. For this study, we call this *Modality-invariant and Class-separable Multimodal AutoEncoder (MCMAE)*.

## 3.3 DOMAIN DISCRIMINATOR

MCMAE includes domain invariance and class separability. However, class separability between domains is not fully considered because no knowledge related to the unseen domain is given during training, meaning that the domain bias problem cannot be avoided. Therefore, we propose addition of another approach to address this problem.

We introduce a model $p_\beta(d|\boldsymbol{z})$ that discriminates a domain from the shared representation. If this discriminator can be trained together with MCMAE, then the representation is separable between domains, leading to alleviation of the domain bias problem.

For training $p_\beta(d|\boldsymbol{z})$, we create a dataset with all class-attributes as inputs and corresponding domain variables as labels $\mathcal{D}_{tr_a} = \{(\boldsymbol{a}_{y_c}, d_c)\}_{c=1}^{S+U}$. Since the input of this discriminator is $\boldsymbol{z}$, we should sample it from attributes $\boldsymbol{a}_y$ using the inference model. Here, we use a *reconstructed* representation not only via the inference model of attributes but also via the generative and inference model of images. Therefore, the objective of the domain discriminator given $(\boldsymbol{a}_y, d)$ is

$$
\mathcal{L}_{\theta_{x,a},\phi_a,\beta}^D(\boldsymbol{a}_y, d) = E_{\boldsymbol{z}' \sim q_{\phi_x}(\boldsymbol{z}|\boldsymbol{x}), \boldsymbol{x} \sim p_{\theta_x}(\boldsymbol{x}|\boldsymbol{z}), \boldsymbol{z} \sim q_{\phi_a}(\boldsymbol{z}|\boldsymbol{a}_y)}[\log p_\beta(d|\boldsymbol{z}')].
\tag{7}
$$

We will explain the reason for using the reconstructed representation and the details of Eq. 7 in Appendix B. When testing, the domain discriminator must be able to correctly discriminate the domain given an image. It is noteworthy that, for this to work properly, different modalities need to be embedded in the same place in the shared space by each inference model.

By training the objective of the domain discriminator together with that of MCMAE, the embedded shared representation is prompted to be separated by domain. The resulting objective, given $\mathcal{D}_{tr}$ and $\mathcal{D}_{tr_a}$, becomes

$$
\frac{1}{|\mathcal{D}_{tr}|} \sum_{(\boldsymbol{x},\boldsymbol{a}_y) \in \mathcal{D}_{tr}} \mathcal{L}_{\theta_{x,a},\phi_{x,a}}^{MCMAE}(\boldsymbol{x},\boldsymbol{a}_y) + \alpha \cdot \frac{1}{|\mathcal{D}_{tr_a}|} \sum_{(\boldsymbol{a}_y,d) \in \mathcal{D}_{tr_a}} \mathcal{L}_{\theta_{x,a},\phi_a,\beta}^D(\boldsymbol{a}_y, d),
\tag{8}
$$

where $\alpha$ is a hyper-parameter representing the degree to which domain separation is enforced in the representation. We learn the model end-to-end by maximizing Eq.8 in terms of all parameters.

After training, this discriminator can predict the possibility of assigning a domain from any input $\boldsymbol{z}$. In addition, if modality invariant embedding is possible, then one can predict the domain for unseen images and can thereby obtain "prior knowledge of the domain" from this classifier. We propose combination of this with an attribute-based classifier (Eq. 1) to compensate the class prediction as a mixture model:

$$
p(y|\boldsymbol{x}) = p(y|\boldsymbol{x}, d=0)p(d=0|\boldsymbol{x}) + p(y|\boldsymbol{x}, d=1)p(d=1|\boldsymbol{x}),
\tag{9}
$$

where $p(y|\boldsymbol{x}, d=0) = \frac{\exp(F(\boldsymbol{x},\boldsymbol{a}_y))}{\sum_{\hat{y} \in \mathcal{Y}_s} \exp(F(\boldsymbol{x},\boldsymbol{a}_{\hat{y}}))}$, $p(y|\boldsymbol{x}, d=1) = \frac{\exp(F(\boldsymbol{x},\boldsymbol{a}_y))}{\sum_{\hat{y} \in \mathcal{Y}_u} \exp(F(\boldsymbol{x},\boldsymbol{a}_{\hat{y}}))}$, and $p(d=1|\boldsymbol{x}) = \exp(E_{\boldsymbol{z} \sim q_{\phi_x}(\boldsymbol{z}|\boldsymbol{x})}[\log p_\beta(d=1|\boldsymbol{z})])$.

We expect that the domain bias problem can be alleviated by selecting a class that maximizes this probability as the prediction result. We call this approach as *MCMAE with the Domain discriminator (MCMAE-D)*. Figure 2(c) shows the overall architecture of MCMAE-D.

## 4 RELATED WORK

### 4.1 ZERO-SHOT LEARNING / GENERALIZED ZERO-SHOT LEARNING

The main approach for tackling ZSL is to learn a compatibility function that represents the relation between images and class-attributes from the seen data[1]. Some early studies used classifiers such as SVM to learn a mapping from images to class-attributes (Lampert et al., 2009; 2014), but many other methods use linear embedding into attributes (or semantic representations) and typically set a ranking loss as the objective function to train such embedding (Frome et al., 2013; Akata et al., 2015; Romera-Paredes & Torr, 2015; Kodirov et al., 2017). Romera-Paredes & Torr (2015) add the regularization term of the mapping to this objective function, and Kodirov et al. (2017) minimizes the embedding error in both directions with the objective. The nonlinear model is also used (Socher et al., 2013) to take non-linearity between images and attributes into consideration. However, with this mapping direction, a hubness problem occurs in which all images are assigned to one type of attributes. To avoid this, some methods measure the similarity in the image space by obtaining the reverse mapping of attributes to images (Zhang et al., 2017; Verma & Rai, 2017).

Another approach is to map images and attributes to another shared representation space and measure the similarity in that space (Zhang & Saligrama, 2015; 2016), which is adopted in our paper. The advantage of this is that it can obtain a representation that does not depend on the dimension or representation of the input. Moreover, arbitrary representation can be learned by adding constraints during training. Recently, several methods use VAEs to learn this, which we will discuss in Sec. 4.2.

These relation-based methods have the advantage of being able to perform classification just by learning the compatibility function, but they do not work well in GZSL due to the domain bias problem (Chao et al., 2016). One way to alleviate this problem is to calibrate the compatibility function (Chao et al., 2016; Liu et al., 2018). In recent years, synthesis-based methods[2] have become mainstream because of their significant improvement in GZSL performance (Mishra et al., 2017; Verma et al., 2018; Xian et al., 2018b; Felix et al., 2018). In particular, Zhang & Koniusz (2018a) propose a model selection mechanism that improves synthesis-based performance by distinguishing seen and unseen domains from the generated images. However, these synthesis-based methods require costly image generation and additional classifier learning. In this study, we aim to achieve high performance with the end-to-end relation-based method.

### 4.2 MULTIMODAL DEEP GENERATIVE MODELS FOR SHARED REPRESENTATION LEARNING

It has been discussed that the importance of learning appropriate representation in ZSL. Jiang et al. (2017) propose to learn an attribute representation that is discriminative between classes while maintaining their semantic information. In this study, we discuss how to learn a good shared representation that integrates different modalities of images and attributes in ZSL.

In recent years, several methods have been proposed to learn the shared representation with VAEs. JMVAE (Suzuki et al., 2016) and TrELBO (Vedantam et al., 2017) are aimed at obtaining a modality-invariant representation by learning the inference model of each modality close to that of taking two modality inputs. However, because an additional inference model is required, it is expensive in terms of the number of parameters. In addition, regularization that forces the representation to be close to the standard Gaussian prior makes it difficult to obtain a class-separable representation (Jiao et al., 2019).

PSE (Jiao et al., 2019) and CADA-VAE (Schonfeld et al., 2019) are very similar to MCMAE in terms of shared representation learning. PSE is not for ZSL but is equivalent to Eq. 5 by considering label information as attributes $a$ [3]. Their mutual difference is therefore whether they contain the reconstruction term given attributes, imposing a modality invariant representation. CADA-VAE, a

---

[1]Although Xian et al. (2018a) categorizes these methods into four groups, we call all of them relation-based approaches, because all methods learn the relation between images and classes (or attributes) and use it directly to evaluate test data.

[2]In this study, we refer to a method that requires training a classifier given samples generated from a learned model as synthesis-based.

[3]In Jiao et al. (2019), this model is called PSE*, and the model corresponding to Eq. 4 is called PSE.

synthesis-based model on the shared representation, has an objective that consists of three parts[4]: VAE losses for each modality, cross-alignment losses that reconstruct one modality from the other, and distribution-alignment loss that brings the inference models of each modality closer. The main difference between MCMAE is that the inference model is forced to stay close to the prior. Similarly to JMVAE and TrELBO, this might be too restrictive to obtain a good shared representation.

## 5 EXPERIMENTS

### 5.1 DATASET AND SETTING

For experimentation, we use the following four datasets, which are commonly used for ZSL: Animals with Attributes (AWA) (Lampert et al., 2014), CUB-200-2011 Bird (CUB) (Wah et al., 2011), SUN Attribute (SUN) (Patterson & Hays, 2012), and Attribute Pascal and Yahoo (aPY) (Farhadi et al., 2009). Each dataset includes images of each class, where each class is represented by semantic attributes. For fair comparison, we use a 2048-dim top-layer embedding of the 101-layered ResNet (He et al., 2016) provided by Xian et al. (2018a) as the image vector. Furthermore, for the class-attribute representation, we use the attributes valued continuously between 0 and 1 provided with each dataset. We followed the split proposed in Xian et al. (2018a) for splitting each dataset into train, validation, and test. The hyper-parameter selection of our model was based on this train–validation split. In training of GZSL, we used both as training data.

As the metric of evaluation, we use the average per-class top-1 accuracy on both the seen and unseen classes (referred as $acc_s$ and $acc_u$). In addition, to evaluate the performance on GZSL, we calculate the harmonic mean of $acc_s$ and $acc_u$, which is $acc_H = (2 \cdot acc_s \cdot acc_u)/(acc_s + acc_u)$.

The architecture of each deep probabilistic model is listed in Appendix A. We set the dimension of the latent variable to 512 and $\alpha = 0.01$. We used the Adam optimization algorithm (Kingma & Ba, 2014) with a learning rate of $10^{-3}$. In all experiments, we trained for 100 epochs[5]. All models in this paper were implemented using PyTorch (Paszke et al., 2017).

### 5.2 ANALYSIS OF OUR PROPOSED METHOD

This section presents analyses of the proposed method from various perspectives. We use AWA for this analysis because the number of data per class is greater than that for other datasets.

First, we confirm the effectiveness of the proposed method by comparing it with a similar shared representation learning model: PSE and CADA-VAE. To align the conditions, we used the same network structure for the inference and generative models of each modality throughout all methods. The original CADA-VAE paper takes a synthesis-based approach in which samples are generated from the shared space and use for learning a classifier. Here, to compare the effectiveness of shared representation learning, we evaluate CADA-VAE with a relation-based approach similar to MC-MAE. In all methods, the classification probability was obtained by Eq. 1.

Table 1 presents results in GZSL. First, compared with PSE and MCMAE, $acc_s$ is higher in the MCMAE, but $acc_u$ is higher in the PSE. One might infer that this result means that MCMAE obtains a shared representation that is not generalized. However, when learning these models and domain classifiers simultaneously and when classifying them with the mixture prediction model (shown as PSE-D and MCMAE-D), MCMAE-D has markedly higher performance. Furthermore, the performance of the domain discriminator by AUROC is the highest in MCMAE-D. As described in Sec. 3.3, this domain discriminator approach does not work properly unless the modality invariance is generalized. This suggests that MCMAE can obtain a modality-invariant representation. On the other hand, because PSE does not consider modality invariance, the domain discriminator does not generalize well to the image modality. In addition, the result that $acc_u$ is greatly improved by MCMAE-D implies that MCMAE can acquire the class separability representation but its performance is degraded due to the domain bias problem.

---

[4]See Appendix C for details of CADA-VAE.

[5]This was determined based on performance when learning with the training-test split in AWA. For detail of learning progress in our method, see Appendix D.

Table 1: Comparison with shared representation learning models in a GZSL setting. We include the results of combining each model with a domain discriminator (denoted as x-D). AUROC shows the evaluation of domain prediction by the domain discriminator using the area under the curve of the receiver operating characteristic.

| Models | $acc_u$ | $acc_s$ | $acc_H$ | AUROC |
|---|---|---|---|---|
| PSE | 34.8 | 86.4 | 49.6 | - |
| CADA-VAE (relation-based) | 21.4 | 73.9 | 33.1 | - |
| MCMAE | 25.3 | 88.4 | 39.3 | - |
| PSE-D | 36.6 | 58.4 | 45.0 | 0.78 |
| CADA-VAE-D (relation-based) | 50.6 | 43.6 | 46.8 | 0.77 |
| MCMAE-D | 60.4 | 67.9 | 63.9 | 0.89 |

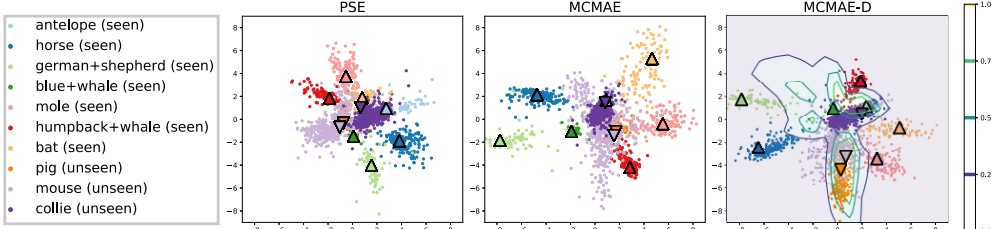

Figure 3: 2-D representation of PSE, MCMAE, and MCMAE-D. These were obtained by learning by setting the $z$ dimension of the model to 2. Circle plots show embedding of the test images by $p_{\phi_x}(z|x)$. Triangles represent embedding of the class-attributes: $\triangle$ denotes a seen class and $\triangledown$ is an unseen one. The contour lines in MCMAE-D represent the domain prediction probability $p(d = 1|z)$ obtained by the domain discriminator.

To confirm this consideration qualitatively, we visualize the shared representation of PSE, MCMAE, and MCMAE-D in two dimensions. Figure 3 portrays the visualization results. In PSE, the test images of the seen classes are well embedded around the corresponding attribute. However, for the unseen class, the data of some classes such as "collie", are embedded in a location that differs from the corresponding attribute, which indicates that the modality invariance is generalized insufficiently. By contrast, in MCMAE, we can confirm that the unseen images are embedded almost appropriately, i.e., we can obtain a generalized modality-invariant representation. However, it is apparent that some class attributes overlap, which can cause the domain bias problem. Then, moving to the MCMAE-D representation, we confirm that embedding of seen and unseen is separated slightly because of learn embedding with the domain discriminator jointly. Furthermore, the domain classifier has prior knowledge to predict which domain is the location before the test images are given. Then we confirm that the test images are actually embedded almost in accordance with it. This demonstrates that the domain discriminator has the ability to alleviate the domain bias problem.

Let us go back to Table 1 and see the CADA-VAE results. Schonfeld et al. (2019) reports that this model performs well with a synthesis-based approach in latent space. However, this result demonstrates that it is worse than PSE and MCMAE in the relation-based case. This is probably true because the representation is over-constrained by the regularization term of the inference model included in CADA-VAE.

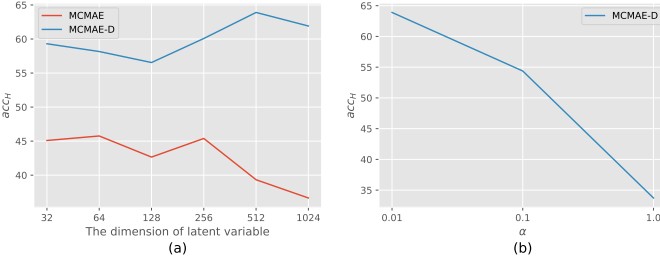

Figure 4: Transition of GZSL performance (the harmonic mean) when each parameter is changed: (a) the dimension of the shared representation and (b) the coefficient of domain discriminator $\alpha$ in MCMAE-D.

Table 2: Comparison with GZSL state-of-the-art models. Bold typeface denotes the best performance among the relation-based models.

| | CUB | | | AWA | | | SUN | | | aPY | | |
|---|---|---|---|---|---|---|---|---|---|---|---|---|
| | $acc_u$ | $acc_s$ | $acc_H$ | $acc_u$ | $acc_s$ | $acc_H$ | $acc_u$ | $acc_s$ | $acc_H$ | $acc_u$ | $acc_s$ | $acc_H$ |
| **Relation-based** | | | | | | | | | | | | |
| GFZSL (Verma & Rai, 2017) | 0.0 | 45.7 | 0.0 | 1.8 | 80.3 | 3.5 | 0.0 | 39.6 | 0.0 | 0.0 | 83.3 | 0.0 |
| CONSE (Szegedy et al., 2015) | 1.6 | 72.2 | 3.1 | 0.4 | **88.6** | 0.8 | 6.8 | 39.9 | 11.6 | 0.0 | **91.2** | 0.0 |
| CMT (Socher et al., 2013) | 7.2 | 49.8 | 12.6 | 0.9 | 87.6 | 1.8 | 8.1 | 21.8 | 11.8 | 1.4 | 85.2 | 2.8 |
| SYNC (Changpinyo et al., 2016) | 11.5 | 70.9 | 19.8 | 8.9 | 87.3 | 16.2 | 7.9 | **43.3** | 13.4 | 7.4 | 66.3 | 13.3 |
| ESZSL (Romera-Paredes & Torr, 2015) | 12.6 | **63.8** | 21.0 | 6.6 | 75.6 | 12.1 | 11.0 | 27.9 | 15.8 | 2.4 | 70.1 | 4.6 |
| SJE (Akata et al., 2015) | 23.5 | 59.2 | 33.6 | 11.3 | 74.6 | 19.6 | 14.7 | 30.5 | 19.8 | 3.7 | 55.7 | 6.9 |
| DEVISE (Frome et al., 2013) | 23.8 | 53.0 | 32.8 | 13.4 | 68.7 | 22.4 | 16.9 | 27.4 | 20.9 | 4.9 | 76.9 | 9.2 |
| ZSKL (Zhang & Koniusz, 2018b) | 21.6 | 52.8 | 30.6 | 17.9 | 82.2 | 29.4 | 20.1 | 31.4 | 24.5 | 10.5 | 76.2 | 18.5 |
| MCMAE | 30.9 | 64.9 | 41.9 | 25.3 | 88.4 | 39.3 | 21.1 | 39.8 | 27.6 | 5.2 | 87.9 | 9.8 |
| MCMAE-D | **51.0** | 38.3 | **43.7** | **60.4** | 67.9 | **63.9** | **47.1** | 28.8 | **35.8** | **20.8** | 52.7 | **29.8** |
| **Synthesis-based** | | | | | | | | | | | | |
| CVAE-ZSL (Mishra et al., 2017) | - | - | 34.5 | - | - | 47.2 | - | - | 26.7 | - | - | - |
| SE-GZSL (Verma et al., 2018) | 41.5 | 53.3 | 46.7 | 56.3 | 67.8 | 61.5 | 40.9 | 30.5 | 34.9 | - | - | - |
| F-CLSWGAN (Xian et al., 2018b) | 43.7 | 57.7 | 49.7 | 59.7 | 61.4 | 59.6 | 42.6 | 36.6 | 39.4 | - | - | - |
| Cycle-(U)WGAN (Felix et al., 2018) | 47.9 | 59.3 | 53.0 | 59.6 | 63.4 | 59.8 | 47.2 | 33.8 | 39.4 | - | - | - |
| CADA-VAE (Schonfeld et al., 2019) | 51.6 | 53.5 | 52.4 | 57.3 | 72.8 | 64.1 | 47.2 | 35.7 | 40.6 | - | - | - |
| ModelSel-3Way (Zhang & Koniusz, 2018a) | - | - | - | 52.6 | 76.7 | 62.4 | - | - | - | 28.4 | 75.5 | 41.2 |

Next, we analyze the parameter sensitivity of our proposed models: The dimension of the shared representation (in both MCMAE and MCMAE-D) and the coefficient of domain discriminator $\alpha$ (in MCMAE-D). Figure 4 presents the results. First, from Figure 4(a), we see that MCMAE performance decreases slightly as the dimension of latent variables increases. This seems to be because in a large dimension, embedding of the unseen domain becomes more difficult. On the other hand, we find that MCMAE-D is robust to the dimension of latent variables. This result shows that the domain classification probability contributes to the compensation of the performance significantly. Next, from Figure 4(b), we find that increasing the value of the coefficient $\alpha$ decreases the GZSL performance because over-enforced separation might cause imperfect embedding from the input.

### 5.3 COMPARISON WITH GZSL STATE-OF-THE-ART MODELS

Table 2 presents the respective performance results obtained for the proposed method and GZSL state-of-the-art methods. First, MCMAE can be seen to have the same performance as existing relation-based models. These models have low accuracy for the unseen domain, indicating that the domain bias problem occurs. Next, in synthetic-based models, the accuracy of the unseen domain is almost identical to that of the seen, indicating that the domain bias problem is mitigated. However, as described above, this approach must generate many images for even unseen classes after training their generative model and must prepare and learn an additional classifier.

Finally, the results obtained with MCMAE-D, which learns domain discriminator jointly, clarify that the domain bias problem has been relaxed greatly and MCMAE-D achieves the best performance among the relation-based models. Additionally, it is apparent that it performs as well as state-of-the-art synthesis-based models.

These results revealed that the domain bias problem in GZSL can be alleviated and that high performance can be achieved using the domain discriminator on the shared representation, even in relation-based approach.

## 6 CONCLUSION

This study addressed the domain bias problem in GZSL. First, we considered domain invariance and class separability as requirements necessary for high performance in ZSL, and introduced MCMAE that learns embedding in a shared representation that satisfy these requirements. Next, we assumed that the domain bias problem occurs when the unseen domain overlaps with the seen domain and, to address this, proposed MCMAE-D to learn the discriminator that distinguishes two domains from its representation. This discriminator not only encourages domain embedding to be separated in the representation space during training, but it also gives prior knowledge of which domain a point in the representation belongs to after training. Therefore, we combined this discriminator and relation-based classification on MCMAE shared representation as a mixture model to ascertain the classification probability. Through experimentation, we confirmed that this approach mitigates the domain bias problem considerably. Future studies will assess application of our model to larger amounts of data while taking advantage of the relation-based method.

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

## A   NETWORK ARCHITECTURES

The Gaussian distribution is parameterized as

$$p(\boldsymbol{z}|\boldsymbol{x}) = \mathcal{N}(\boldsymbol{z}; \boldsymbol{\mu}, \boldsymbol{\sigma}^2), \boldsymbol{\mu} = f_\mu(f_{\mathrm{MLP}}(\boldsymbol{x})), \boldsymbol{\sigma} = \mathrm{Softplus}(f_\sigma(f_{\mathrm{MLP}}(\boldsymbol{x}))),$$

where $f_{\mathrm{x}}$ represents a neural network and $\mathrm{Softplus}$ is a softplus function.

Moreover, the Laplace distribution with a constant scale $c$ is parameterized as follows.

$$p(\boldsymbol{x}|\boldsymbol{z}) = \mathcal{L}(\boldsymbol{x}; \boldsymbol{\mu}, 2c^2), \boldsymbol{\mu} = f_\mu(f_{\mathrm{MLP}}(\boldsymbol{z})).$$

In this paper, it is assumed that only the mean parameter $\boldsymbol{\mu}$ is output deterministically when sampling from the Laplace distribution with a constant scale:

$$\boldsymbol{x} \sim \mathcal{L}(\boldsymbol{x}; \boldsymbol{\mu}, 2c^2) \Leftrightarrow \boldsymbol{x} = \boldsymbol{\mu}.$$

For the notation of model structures, we denote a linear fully-connected layer with $k$ units as `k`, batch normalization, and ReLU as `DkBR`. Also, we denote `DkBR` without batch normalization and ReLU as `Dk`. In addition, the process of applying `J` after `I` is denoted as `I-J`, and the process of concatenating the last layers of the two networks `I, J` into one layer is denoted as `(I, J)`.

The network structures of distributions of MCMAE are as follows (`DdimA` is the dimension of attributes and `DdimZ` is that of latent variable):

- $p_{\theta_x}(\boldsymbol{x}|\boldsymbol{z})$ (Laplace)
    - $f_\mu$: `D2048`
    - $f_{\mathrm{MLP}}$: `z-D1024BR-D1024BR`
- $p_{\theta_a}(\boldsymbol{a}|\boldsymbol{z})$ (Laplace)
    - $f_\mu$: `DdimA`
    - $f_{\mathrm{MLP}}$: `z-D1024BR-D1024BR`
- $q_{\phi_x}(\boldsymbol{z}|\boldsymbol{x})$ (Gaussian)
    - $f_\mu$ and $f_{\sigma^2}$: `DdimZ`
    - $f_{\mathrm{MLP}}$: `x-D1024BR-D1024BR`
- $q_{\phi_a}(\boldsymbol{z}|\boldsymbol{a})$ (Gaussian)
    - $f_\mu$ and $f_{\sigma^2}$: `DdimZ`
    - $f_{\mathrm{MLP}}$: `a-D1024BR-D1024BR`

Moreover, the structure of the domain discriminator $p_\lambda(d|\boldsymbol{z}) = \mathcal{B}(d; \mu = \mathrm{Sigmoid}(f_\mu(f_{\mathrm{MLP}}(\boldsymbol{z}))))$ (where $\mathcal{B}$ means Bernoulli distribution and $\mathrm{Sigmoid}$ is a sigmoid function) is as follows.

- $p_\lambda(d|\boldsymbol{z})$ (Bernoulli)
    - $f_\mu$: `D1`
    - $f_{\mathrm{MLP}}$: `z-D1024BR`

## B   LEARNING THE DOMAIN DISCRIMINATOR

A straightforward way to learn the domain discriminator is to use an inference model $q_{\phi_a}(\boldsymbol{z}|\boldsymbol{a}_y)$ to maximize the following objective function:

$$E_{\boldsymbol{z} \sim q_{\phi_a}(\boldsymbol{z}|\boldsymbol{a}_y)}[\log p_\beta(d|\boldsymbol{z})]. \tag{10}$$

By learning both the inference model and the discriminator together, we expect to obtain a representation that can be separated by the domain. However, only $q_{\phi_a}(\boldsymbol{z}|\boldsymbol{a}_y)$ will be affected strongly by the learning of the discriminator and might not be stable during training. In addition, the representation that we want to discriminate in the test time is embedded from the images, not from the attributes. Therefore, we take a method of generating image data in pseudo and learning the discriminator

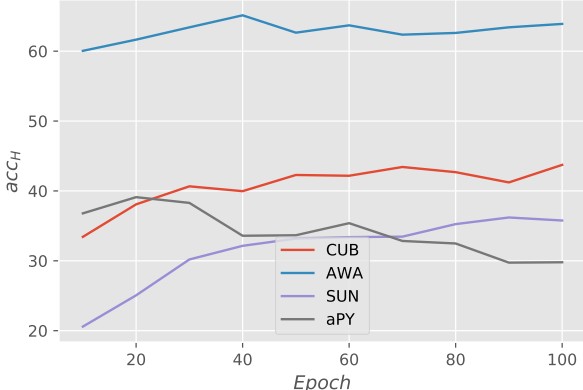

Figure 5: Learning curves of MCMAE-D for each dataset.

on the representation inferred from them, which is why we use the reconstructed representation in discriminator learning.

Because attributes have less variation than images, the variance of representation sampled from $q_{\phi_a}(\boldsymbol{z}|\boldsymbol{a}_y)$ becomes small, and as a result, that of the reconstructed representation can also be small, which may lead to overfitting of the domain discriminator on the representation during training. To alleviate this, we consider increasing the variance of the inference model when calculating Eq.7, that is, we consider the inference model as $q_{\phi_a}(\boldsymbol{z}|\boldsymbol{a}_y) = \mathcal{N}(\boldsymbol{z}; \boldsymbol{\mu}, a^2\boldsymbol{\sigma}^2)$, where $a > 1$. In this study, we set $a = 2$. Note that this change is only for Eq.7 and does not apply to other calculations such as the objective function of MCMAE (Eq.6).

## C    CADA-VAE

The objective of CADA-VAE (Schonfeld et al., 2019) is as follows:

$$
\begin{aligned}
\mathcal{L}^{CADA-VAE}_{\theta_{x,a},\phi_{x,a}}(\boldsymbol{x}, \boldsymbol{a}_y) &= \mathcal{L}^{VAE}_{\theta_x,\phi_x}(\boldsymbol{x}) + \mathcal{L}^{VAE}_{\theta_a,\phi_a}(\boldsymbol{a}_y) \\
&\quad + \gamma(E_{q_{\phi_a}(\boldsymbol{z}|\boldsymbol{a})}[\log p_{\theta_x}(\boldsymbol{x}|\boldsymbol{z})] + E_{q_{\phi_x}(\boldsymbol{z}|\boldsymbol{x})}[\log p_{\theta_a}(\boldsymbol{a}|\boldsymbol{z})]) \\
&\quad - \delta W_2(q_{\phi_x}(\boldsymbol{z}|\boldsymbol{x}), q_{\phi_a}(\boldsymbol{z}|\boldsymbol{a}_y)),
\end{aligned}
\tag{11}
$$

where $W_2(p, q)$ is the 2-Wasserstein distance between $p$ and $q$ and where $\delta$ and $\gamma$ are weighting factors. Moreover, $\mathcal{L}_{VAE_a}$ is the objective of VAE that takes attributes as inputs and uses inference and generative models for attributes.

In our experiment on Table 1, all coefficients in Eq. 11 are set to 1 for a fair comparison with MCMAE. Moreover, in the relation-based setup, we used the Wasserstein distance instead of KL divergence to calculate the compatibility function (Eq. 2).

## D    LEARNING PROGRESS OF MCMAE-D

One advantage of relation-based methods including MCMAE-D is that GZSL performance can be verified during model learning. On the other hand, the synthesis-based method cannot be confirmed it unless learning of the generative model is completed and the synthesis data are generated. Figure 5 shows learning curves of MCMAE-D for each dataset. From this result, it can be seen that the learning progress of this model is almost stable, although the performance of aPY drops before 100 epochs.

