# OpenReview forum: "Relation-based Generalized Zero-shot Classification with the Domain Discriminator on the shared representation"
_ICLR.cc/2020/Conference — Reject_

### Official Review · AnonReviewer1 · 2019-10-18
**Official Blind Review #1**

**Rating:** 6

**Review:**

The paper presents a novel approach for (generalized) Zero-shot learning (GZSL). As showing in the numerical experiments on some real data, the method demonstrates the significant improvement on the accuracy of prediction comparing to some state-of-the-art methods.
The main key of the method is using Variational Inference, variational autoencoders. The authors have taken into account the modality of the data through reparametrize the distributions, especially the inside class invariant modality and class separability. Moreover, the authors also propose to take into account a kind of biasness domain into the learning procedure, which details in adding a regularization of the domain discriminator into the objective function.

The paper is nicely written, espcially with a clear formal introduction to the problem of GZSL.

However, I have some questions:
1) Does the test set has some labels? How do you know your method works well? I can not find where you have defined a kind of loss so that we can compare the predicted labels \hat{y}_j ?  (In Section 2.)
2) How do you learn the "replaced prior" in equation (4) ?
3) It is not enough detail on how do you optimize the objective (8) ? a detail explain algorithm would make the paper significant, indeed.
4) In Table1, would MCMVAE in the last row be MCMVAE-D ?

Final, I expect the authors will make their codes available for the readers.

**Experience Assessment:**

I have published one or two papers in this area.

**Review Assessment: Checking Correctness Of Derivations And Theory:**

I carefully checked the derivations and theory.

**Review Assessment: Checking Correctness Of Experiments:**

I assessed the sensibility of the experiments.

**Review Assessment: Thoroughness In Paper Reading:**

I read the paper at least twice and used my best judgement in assessing the paper.

---

> ### Author Response · Authors · 2019-11-13
> **Response to Review #1**
>
> Thank you very much for providing comments.
> We will answer your questions. We will also upload a revision based on your comments later.
>
> ---
> > 1) Does the test set has some labels? How do you know your method works well? I can not find where you have defined a kind of loss so that we can compare the predicted labels \hat{y}_j ?  (In Section 2.)
>
> Yes, all test set examples have true labels, but of course, they cannot be confirmed during training. To see if the model works well in the test phase, we compare these true labels with the labels predicted in Eq.1. In the experiment of section 5.2, the performance of the model is evaluated by the average per-class top-1 accuracy on both the seen and unseen classes, and the harmonic mean of them. As pointed out, it is a little difficult to understand for now, so I will revise it later.
>
>
> > 2) How do you learn the "replaced prior" in equation (4) ?
>
> Like other parameterized probability distributions, $q_{\phi_{a}}(z|a_y)$ is learned by maximizing the objective function with respect to $\phi_{a}$.
>
>
> > 3) It is not enough detail on how do you optimize the objective (8) ? a detail explain algorithm would make the paper significant, indeed.
>
> We learn all models end-to-end by maximizing Equation 8 for all these parameters $\theta_{x,a}, \phi_{x,a}, \beta$. We didn't write about this, so we'll add it later.
> For optimization, we used Adam optimizer, which is described in section 5.1.
>
>
> > 4) In Table1, would MCMVAE in the last row be MCMVAE-D ?
>
> This is as you pointed out. I will fix it later.
>
>
> > Final, I expect the authors will make their codes available for the readers.
>
> We are going to share our codes, but we may not be able to release them within the rebuttal period because we need to rearrange them.
> ---
>
> Again, we appreciate all of your comments.

---

### Official Review · AnonReviewer3 · 2019-10-25
**Official Blind Review #3**

**Rating:** 3

**Review:**

Summary:
This paper proposes a relation-based ZSL model which can effectively alleviate the domain bias problem. To this end, first, the paper claims that a good relation-based ZSL model should consider two requirements -- modality invariance and class separability. And the paper designed Modality-invariant and Class-separable Multimodal VAE (MCMVAE) based on VAEs to meet the two aforementioned requirements. Next, the paper hypothesizes that the domain bias problem is due to the overlap between seen and unseen classes in the shared space, and explicitly introduced a discriminator to separate the two domains. The paper performs experiments on ZSL benchmark datasets and shows that the proposed method outperforms other relation-based methods. Besides, the domain discriminator which can be applied to other models demonstrates its effectiveness in reducing domain bias given the experimental results.

+Strengths:
1. Clear writing logic. The author clearly depicts how to get the final loss of the method step-to-step and the relationship with existing methods.
2. The version without the domain discriminator (i.e. MCMVAE) is similar to PSE and CADA-VAE as the author acknowledges. However, the domain discriminator has certain novelty and can be applied to other methods. The overlap among seen and unseen classes is an important problem (domain bias problem named by the author) and the add of the domain discriminator to distinguish whether a sample is from seen classes or unseen classes is reasonable, which can provide better class separability (among seen and unseen classes).

-Weaknesses:
1. Although the author claims that the proposed method is a relation-based method, it is strange that the proposed method is called xxVAE but in Table 2 it doesn't fall into synthesis-based methods (as CVAE-ZSL and CADA-VAE do). Although it is derived from VAE, the current method doesn't seem to be called a VAE any more (some of the regularizations of the VAE are relaxed). Also, are the two terms -- relation-based and synthesis-based -- first proposed by the author? Is there a clear boundary between those two groups of methods?
2. It is recommended that an additional figure that depicts the framework is added (similar to Figure 2 in CADA-VAE) to promote better understanding. Currently, the method part only contains formulas with many parameters, making it difficult to grasp the idea of the whole framework at first glance.
3. The novelty of this paper is somewhat limited while missing some relevant works, e.g.[r1, r2]. [r1] learns a latent space where the compactness within class and separateness between classes are considered. [r2] uses a two-stage prediction for GZSL.
[r1] Jiang et al. Learning Discriminative Latent Attributes for Zero-Shot Classification. In IEEE ICCV 2017.
[r2] Zhang et al. Model Selection for Generalized Zero-shot Learning. In arXiv 2018.
4. It is a question whether the seen and unseen classes can be separated (Whether a two stage process is correct?). The key for ZSL is knowledge transfer and the base is that seen and unseen classes are related [r3]. If they are separated, can one use the model trained on seen classes to recognize the unseen classes? This is quite problematic. Besides, in Tab.2 there lacks of necessary comparisons with recent relation-based approaches e.g.[r3][r4], which makes the evaluation less sufficient.
[r3] Jiang et al. Transferable Contrastive Network for Generalized Zero-Shot Learning. In IEEE ICCV 2019.
[r4] Li et al. Discriminative Learning of Latent Features For Zero-Shot Recognition. In IEEE CVPR 2018.
5. Some unclear/incorrect descriptions of the method:
5.1) The formulation of GZSL is incorrect. Y= union(y_s  y_u), but not intersection(y_s  y_u)
5.2) How is the class separation formulated in the framework?
5.3) In Sec.3.2, why is the log-likelihood of the generative models can be obtained by the L1 loss?

Minor issues:
1. Better use vectorgraphs for clear view (especially for Figure 3 and 4).
2. Incomplete reference: for Probabilistic semantic embedding (PSE), the reference should add the conference information.
3. Grammar and spelling mistakes:
[1] Content in Figure 2 (not caption): unseen class -> unseen classes
[2] Last line in 4.1: MCVAE-D -> MCMVAE-D
[3] Last paragraph in 4.2: close -> stay close
[4] Last model name in Table 1: MCMVAE -> MCMVAE-D
4. The color bar for the contours at the rightmost of Figure 3 is not clear (not the standard way to draw a color bar, better refer to what a color bar is usually drawn).
5. If possible, better reduce the main text to 8 pages as recommended by the submission instructions (e.g. some content of the method part can be moved to the appendix?).

**Experience Assessment:**

I have published in this field for several years.

**Review Assessment: Checking Correctness Of Derivations And Theory:**

I carefully checked the derivations and theory.

**Review Assessment: Checking Correctness Of Experiments:**

I carefully checked the experiments.

**Review Assessment: Thoroughness In Paper Reading:**

I read the paper thoroughly.

---

> ### Author Response · Authors · 2019-11-13
> **Response to Review #3 (1/2)**
>
> Thank you very much for providing detailed comments.
> We will answer your questions. In addition, we will upload a revised version reflecting your comments later.
>
> > 1. Although the author claims that the proposed method is a relation-based method, it is strange that the proposed method is called xxVAE but in Table 2 it doesn't fall into synthesis-based methods (as CVAE-ZSL and CADA-VAE do). Although it is derived from VAE, the current method doesn't seem to be called a VAE any more (some of the regularizations of the VAE are relaxed). Also, are the two terms -- relation-based and synthesis-based -- first proposed by the author? Is there a clear boundary between those two groups of methods?
>
> As you pointed out, the terms "relation-based" and "synthesis-based" were proposed by us, but we believe that the difference between them is clear.
> The relation-based method learns a compatibility function during training and uses it to predict labels during testing. This method does not require an explicit classifier but classifies labels according to Eq.1. Therefore, this method has the advantage that it can be easily extended to any number of classes if the compatibility function is generalized to the unseen domain.
> The synthesis-based method, on the other hand, learns a generative model from attributes. However, since it cannot perform classification including unseen classes itself, it is necessary to prepare and learn a classifier that predicts (both seen and unseen) class labels from images generated from the generative model. In other words, synthesis-based is a framework that requires the training of a classifier given the synthesized data.
>
> Therefore, the difference between relation-based and synthesis-based is whether we need to synthesize data from a model and learn a classifier. Hence, it is not always consistent with the use of deep generative models.
> For example, CVAE-ZSL generates images from a decoder after learning VAE and classifies the labels of them using a classifier such as SVM. On the other hand, although our proposed model is a deep generative model, it learns the compatibility function (Eq.2) and makes predictions using it. Therefore, our proposed model does not require both data synthesis and training of another classifier, so it can be regarded as a relation-based approach.
>
> In addition, as you pointed out, the proposed model is no longer a VAE, so we will change the name of the proposed method to MCMAE (Modality-invariant and Class-separable Multimodal AutoEncoder) in the modified version.
>
> > 2. It is recommended that an additional figure that depicts the framework is added (similar to Figure 2 in CADA-VAE) to promote better understanding.
> Currently, the method part only contains formulas with many parameters, making it difficult to grasp the idea of the whole framework at first glance.
>
> I understand. As you pointed out, we will include an additional figure of the framework in the revised version.
>
> >> 3. The novelty of this paper is somewhat limited while missing some relevant works, e.g.[r1, r2]. [r1] learns a latent space where the compactness within class and separateness between classes are considered. [r2] uses a two-stage prediction for GZSL.
>
> Thank you for introducing relevant works. We will refer them to the related work section and discuss the differences with the proposed method. In particular, we would like to list [r2] as a synthesis-based GZSL in Table 2.
>
> > 4. It is a question whether the seen and unseen classes can be separated (Whether a two stage process is correct?). The key for ZSL is knowledge transfer and the base is that seen and unseen classes are related [r3]. If they are separated, can one use the model trained on seen classes to recognize the unseen classes? This is quite problematic. Besides, in Tab.2 there lacks of necessary comparisons with recent relation-based approaches e.g.[r3][r4], which makes the evaluation less sufficient.
>
> We conducted an experiment that we trained MCMAE-D with training data, which was the seen domain, and then classified test data using the domain discriminator. As a result, the domain classification performance of MCMAE-D is higher than that of PSE-D and CADA-VAE-D (relation-based). The evaluation by AUROC is as follows.
>
> PSE-D: 0.78
> CADA-VAE-D (relation-based) : 0.77
> MCMAE-D: 0.89
>
> From this result, it was confirmed that the proposed model can discriminate the unseen classes from the test set (these results will be included in the revised version later).
> Furthermore, from the result of acc_u of MCMAE-D, it can be seen that the unseen classes can be classified with almost the same performance as acc_s.
> Therefore, we can conclude that the representation learning of the proposed method generalizes (transfers) to the unseen domain to some extent.

---

> ### Author Response · Authors · 2019-11-13
> **Response to Review #3 (2/2)**
>
> >>  Besides, in Tab.2 there lacks of necessary comparisons with recent relation-based approaches e.g.[r3][r4], which makes the evaluation less sufficient.
>
> Thank you for presenting related works. [r4] is not the result of GZSL, so we will not list it in Table 2.
> On the other hand, [r3] is a study on relation-based GZSL, which is related to our method. However, I have doubts about the results of this paper. This is because it is not written how many epoch numbers the training was completed at the time of training. We ran the implementation by the authors of this paper[1], but we confirmed that its performance overfits as the number of epochs increases and becomes much lower than the results of the original paper.
> In this implementation, the model in the epoch with the best *test* performance is stored as ```"Best_model_GZSL_H_X_S_X_U_X.tar"`. From this, it is suspected that the accuracy described in this paper selects at the epoch number with the best test accuracy, but it is illegal. Therefore, we have doubts about this result, so we will not compare this in Table 2.
> We are also considering to present the evaluation of the proposed method at the epoch when test performance is best in the appendix for reference.
> [1] http://vipl.ict.ac.cn/resources/codes
>
> > 5.2) How is the class separation formulated in the framework?
> The class separation means the degree to which the representation $z$ inferred by $q (z | x)$ and q $(z | a)$ is separated in the representation space for each class.
>
> > 5.3) In Sec.3.2, why is the log-likelihood of the generative models can be obtained by the L1 loss?
> This was our mistake. To be correct, we use a Laplace distribution with a fixed scale as the generative model. Therefore, the log-likelihood is *negative * absolute-difference loss. When sampling from this generated model, only a mean parameter is output deterministically, as with many VAE methods.
> We will fix these in the revised version later.
>
> > 2. Incomplete reference: for Probabilistic semantic embedding (PSE), the reference should add the conference information.
>
> This research was rejected at ICLR last year and is only available in openreview (not uploaded to arXiv). Therefore, we cannot add conference information, so we will add the openreview link of this paper to the reference instead.
>
> > 5.1) The formulation of GZSL is incorrect. Y= union(y_s  y_u), but not intersection(y_s  y_u)
> > 1. Better use vectorgraphs for clear view (especially for Figure 3 and 4).
> > 3. Grammar and spelling mistakes:
> > 4. The color bar for the contours at the rightmost of Figure 3 is not clear (not the standard way to draw a color bar, better refer to what a color bar is usually drawn).
>
> Thank you. We will fix these later.
>
> >> 5. If possible, better reduce the main text to 8 pages as recommended by the submission instructions (e.g. some content of the method part can be moved to the appendix?).
>
> We think that all explanations of the proposed method are important and it is difficult to move them to the appendix, but in the revised version we will try to reduce the number of pages as much as possible.
>
> Again, we appreciate all of your comments.

---

### Official Review · AnonReviewer2 · 2019-10-30
**Official Blind Review #2**

**Rating:** 6

**Review:**


The main topic of this paper is generalized zero-shot learning. This paper modifies traditional VAE method with attribute matching prior to release the hidden features from original regularization. This paper also proposes a domain discriminator to enhance class-separability of learned features to avoid unseen classes to be covered by seen classes. Experiment results show their efficiency under relation-based setting.

Pros:
1.This paper proposes an important insight that in generalized ZSL, the unseen classes may be dominated by seen classes in the feature space.
2.An easy but efficient domain discriminator method is proposed to separate different classes to avoid domination.
3.Even without large synthetic learning architecture, the proposed method gets comparable results.

Comments:
1.The proposed MCMVAE is No-longer a VAE but an AE with attribute matching loss. Except that a new theory of MCMVAE is proposed, it is not rigorous to relate MCMVAE to VAE.
2.Add results using synthetic architecture to get a better result will make this method more reliable.
3.Why discriminator is harmful for PSE method?


**Experience Assessment:**

I have read many papers in this area.

**Review Assessment: Checking Correctness Of Derivations And Theory:**

I carefully checked the derivations and theory.

**Review Assessment: Checking Correctness Of Experiments:**

I carefully checked the experiments.

**Review Assessment: Thoroughness In Paper Reading:**

I read the paper thoroughly.

---

> ### Author Response · Authors · 2019-11-13
> **Response to Review #2**
>
> We would like to thank the reviewer for providing comments.
> We will answer your questions. In addition, we will upload a revised version reflecting your comments later.
>
> ---
> > 1.The proposed MCMVAE is No-longer a VAE but an AE with attribute matching loss. Except that a new theory of MCMVAE is proposed, it is not rigorous to relate MCMVAE to VAE.
>
>
> MCMVAE is proposed based on VAE, but as you pointed out, it is no longer VAE. Therefore, we will change the name of the proposed method from MCMVAE to MCMAE (Modality-invariant and Class-separable Multimodal AutoEncoder).
>
>
> > 2.Add results using synthetic architecture to get a better result will make this method more reliable.
>
> Although it is an interesting proposal, the purpose of this study is to propose a high-performance model with a relation-based approach. Therefore, I think that building a synthetic-based architecture based on the proposed model is out of the scope of this study.
>
>
> > 3.Why discriminator is harmful for PSE method?
>
> As explained in section 3.3, when learning the domain discriminator, we learn end-to-end inference and generative models in Eq.7 so that representation can be distinguished between domains well.
> At this time, in order for this domain discriminator to work well for test data, different modalities corresponding to the same example need to be embedded in the same latent space by each inference model, that is, they must be modality invariant.
> However, in PSE, there is no term that guarantees modality invariance like MCMAE Eq.6. Therefore, the modality-invariant inference model is not learned in PSE, and as a result, the performance becomes worse due to the correction of the discriminator.
> In addition, we newly verified whether the test data domain can be correctly classified by the domain discriminator. As a result, we confirmed that the proposed method is able to classify the domain most appropriately. This means that the proposed method has obtained the most modality-invariant representation. These results will be added to the revised version later.
> ---
>
> Again, we appreciate all of your comments.

---

### Author Response · Authors · 2019-11-14
**Update our paper**

We thank all reviewers for their insightful and detailed comments. We updated our paper according to your feedback.  In addition to the parts pointed out by the reviewers, we also fixed the following:

- Simplification and clarification of descriptions of related studies and experiments. We also moved some parts (explaining CADA-VAE and domain discriminator learning) to the appendix.

- Fixed some errors and mistakes. In particular, we corrected Eq.8 because it was wrong.

- We redrew some figures, such as Fig.1, to make them easier to see.

We emphasize that these revisions do not harm the overall contribution of this paper.
Moreover, we're sorry for Review #3 but we couldn't reduce the main text to 8 pages, but we reduced it from 10 pages to 9 pages.

Thank you.

---

> ### Author Response · Authors · 2019-11-15
> **Update our paper**
>
> We have corrected some more errors and typographical errors. In particular, the description of the hyperparameter that adjusts the variance of the inference model in the computation of the objective function of the domain discriminator was missing, so we mentioned this in the appendix.
>
> Thank you.

---

### Decision · Program_Chairs · 2019-12-19

**Decision:**

Reject

**Comment:**

This paper proposes a relation-based model that extends VAE to explicitly alleviate the domain bias problem between seen and unseen classes in the setting of generalized zero-shot learning.

Reviewers and AC think that the studied problem is interesting, the reported experimental results are strong, and the writing is clear, but the proposed model and its scientific reasoning for convincing why the proposed method is valuable is somewhat limited. Thus the authors are encouraged to further improve in these directions. In particular:

- The idea of using a variant of the widely-used domain discriminator to make seen and unseen classes distinguishable is somewhat contradicted to the basic principle of zero-shot learning. How to trade off the balance between seen and unseen classes has been an important problem in generalized ZSL. These problems need further elaboration.

- The proposed model itself is not a real "VAE", making the value of an extensive derivation based on variational inference less prominent.

- There is also the need to compare with the baselines mentioned by the reviewers.

Overall, this is a borderline paper. Since the above concerns were not addressed convincingly in the rebuttal, I am leaning towards rejection.